# Phenotypic Variations and Bioactive Constituents among Selected *Ocimum* Species

**DOI:** 10.3390/plants13010064

**Published:** 2023-12-24

**Authors:** Sintayehu Musie Mulugeta, Zsuzsanna Pluhár, Péter Radácsi

**Affiliations:** Department of Medicinal and Aromatic Plants, Institute of Horticultural Sciences, Hungarian University of Agriculture and Life Sciences, Villányi út 29-43, H-1118 Budapest, Hungaryradacsi.peter@uni-mate.hu (P.R.)

**Keywords:** variability, herb yield, essential oil content, essential oil composition

## Abstract

Basil (*Ocimum* species) represents an extraordinary group of aromatic plants that have gained considerable economic importance, primarily due to their essential oils, which have applications in medicine, culinary, and perfumery. The *Ocimum* genus encompasses more than 60 species of herbs and shrubs originally native to tropical regions. This genus stands out for its remarkable diversity, displaying a wide spectrum of variations in phenotype, chemical composition, and genetic makeup. In addition to genetic factors, the growth, development, and essential oil production of basil are also influenced by environmental conditions, ontogeny, and various other factors. Consequently, the primary objective of this study was to explore the diversity in both the morphological characteristics and essential oil composition among basil genotypes preserved within the gene bank of the Hungarian University of Agriculture and Life Sciences’ Department of Medicinal and Aromatic Plants. The investigation involved the assessment of fifteen basil genotypes, representing four distinct species: *Ocimum basilicum* (including ‘Anise’, ‘Clove’, ‘Fino Verde’, ‘Licorice’, ‘Mammoth’, ‘Mrs. Burns’, ‘Thai tömzsi’, ‘Thai hosszú’, and ‘Vietnamese basil’), *Ocimum sanctum* (green holy basils), *Ocimum citrodora* (Lemon basil), and *Ocimum gratissimum* (African and Vana holy basil). The genotypes exhibited significant variations in their morphological growth, essential oil content (EOC), and composition. African basil produced more biomass (408.3 g/plant) and showed robust growth. The sweet basil cultivars clove, licorice, Thai tömzsi, and Thai hosszú also exhibited similar robust growth trends. Vietnamese basil, on the other hand, displayed the lowest fresh biomass of 82.0 g per plant. Both holy basils showed EOC levels below 0.5%, while Mrihani basil stood out with the highest EOC of 1.7%. The predominant constituents of the essential oil among these genotypes comprised estragole, thymol, methyl cinnamate, linalool, and eugenol. In conclusion, this study showed that the genotypes of basil stored in the department’s gene bank exhibit a wide range of variability, both within and between species.

## 1. Introduction 

The genus *Ocimum* L. belongs to the *Lamiaceae* family and is commonly referred to as basil. The name “basil” originates from the Greek word “basilikos”, which translates to “royal”. *Ocimum* species encompass a variety of annual and perennial herbs and shrubs. These plants are distributed across different geographical regions, with three primary areas of *Ocimum* diversity identified: (a) the tropical and subtropical regions of Africa; (b) the tropical parts of Asia; and (c) the tropical areas of America, particularly in Brazil. However, basil is extensively cultivated worldwide [1]. The genus presents extensive morphological diversity, encompassing approximately 65 species due to widespread cultivation and the ease of cross-pollination. This has led to the emergence of numerous subspecies, varieties, and forms [2,3,4]. Among the most well-known species renowned for their strong aromatic properties are *Ocimum basilicum*, *Ocimum gratissimum*, *Ocimum sanctum*, and *Ocimum americanum*. In addition to these species, there are several varieties and related species, or hybrids [3,4]. The genus is recognized for its polymorphism, extensive chemical diversity [5], which varies in both volatile and non-volatile compounds [6,7,8], and a significant level of genetic variability [9,10]. Morphologically, basil can be identified by its square stems, leaves positioned opposite each other on the stem, and the presence of brown or black seeds, known as nutlets, along with flower spikes. Nevertheless, the characteristics of basil leaves and flowers, such as their color, size, shape, and texture, exhibit variations among different species. The leaves exhibit diverse textures, ranging from smooth and glossy to curled and covered in fine hairs. Additionally, basil plants can display a range of leaf colors, from green to shades of blue and purple. Furthermore, the height and canopy spread of basil plants can differ based on their specific species [1,5,11]. On the other hand, the essential oil content varies between 0.2% and 5.22%, primarily influenced by factors such as the plant species, its origin, and its phenological stage [2,12]. Researchers worldwide have extensively examined the chemical compositions of basil essential oil, resulting in the identification of approximately 140 constituents. These components include monoterpenes, sesquiterpenes, carboxylic acids, aliphatic aldehydes, aliphatic alcohols, aromatic compounds, and other miscellaneous compounds [2,13,14]. Among the primary monoterpene derivatives are linalool, camphor, 1,8-cineole, thymol, citral, and geraniol. In contrast, other members of the same genus typically possess essential oils characterized by significant quantities of phenolic derivatives such as eugenol, methyl eugenol, methyl chavicol (estragole), and methyl cinnamate, often accompanied by varying levels of linalool [2,14,15]. Apart from volatile essential oils, basil herbs are also rich in polyphenols ranging from phenolic acids (rosmarinic acid, caffeic acid, caftaric acid, chicoric acid, and others), simple or complex flavonoids, and colored anthocyanins [16,17,18]. The aromatic essential oils and the polyphenols of basil species are used in the flavor, fragrance, cosmetics, aromatherapy, and pharmaceutical industries [13,19]. They exhibit antioxidant, anti-inflammatory, antimicrobial, and immunomodulatory activities [20,21,22]. In addition, the essential oils have demonstrated insecticidal and repellent properties, making them useful in pest control [23,24,25]. The morphological development, essential oil content, and composition of *Ocimum* species are significantly influenced by genetic factors, growing conditions, phenology, and other elements [26,27,28]. Consequently, this study aims to explore the variations in both the phenotypic morphological characteristics and essential oil composition among 15 basil genotypes cultivated in Budapest, Hungary, in 2022.

## 2. Results 

### 2.1. Morphological Variability

The 15 different *Ocimum* genotypes displayed significant morphological differences, as indicated in Table 1 and Table 2 (*p* < 0.01). These variations were observed across a range of morphological characteristics, including plant height, canopy diameter, leaf size and weight, inflorescence length and number, and biomass production (both fresh and dry). In terms of height, *Ocimum* species ranged from low-growing plants of ‘Fino Verde’, which had a height of 23.0 cm, to taller plants of ‘African basil’ reaching 56.3 cm in height. Sweet basil cultivars such as ‘Fahéj illatú’, ‘Licorice’, ‘Thai tömzsi’, and ‘Thai hosszú’ also had over 50.0 cm taller plants. The canopy morphology also varies greatly, with some species having dense and bushy canopies, such as ‘Vana holy basil’ (27.2 cm) and ‘Fino Verde’ (32.5 cm), and others having more open and spreading canopies, such as ‘Anise’, ‘Clove’, ‘Licorice’, and ‘Thai hosszú’, which had over 50.5 cm spread. Furthermore, leaf morphology shows significant diversity among the *Ocimum* genotypes. The sweet basil cultivar ‘Mammoth’ stands out with its exceptionally long leaves, measuring 12.4 cm in length and 8.6 cm in width, with a larger area of 52.0 cm^2^ per leaf and weighing 214.0 g per 100 leaves. African basil exhibited the second largest leaves, measuring 10.3 cm in length and 6.9 cm in width, covering an area of 36.8 cm^2^ per leaf, and weighing 148 g per 100 leaves. In contrast, ‘Fino Verde’ exhibits smaller leaf growth, with leaves that are 1.8 cm longer and 0.9 cm wider, covering an area of 1.4 cm^2^ per leaf, and weighing 3.5 g less per 100 leaves. The remaining genotypes fall within these ranges.

Inflorescence is another trait that varies among *Ocimum* species. Fahéj illatú, ‘Licorice’, ‘Mrs. Burns’, and ‘Thai hosszú’ displayed inflorescence lengths exceeding 25 cm. In contrast, ‘Fino Verde’ had the shortest inflorescences, measuring only 6.5 cm, but it had the highest inflorescence count (152.7). Variability in biomass production was noticeable among the different genotypes, resulting in a range of growth and yield outcomes (see Table 1 and Table 2). In line with that, sweet basil cultivars such as ‘Clove’, ‘Mammoth’, and ‘Thai tömzsi’, as well as ‘African basil’, yielded over 395 g/plant of fresh herb yield and 80 g/plant of dry herb yield per plant. In contrast, ‘Vietnamese basil’ and ‘Vana holy basil’ produced significantly lower yields, with less than 100 g of fresh herb yield and 25 g of dry herb yield per plant.

Furthermore, through hierarchical cluster analysis, we were able to identify four distinct morphological categories, as depicted in Figure 1. The initial cluster consisted of five basil genotypes, namely ‘Anise’, ‘Mrihani’, ‘Fino Verde’, ‘Lemon’, and ‘green holy basil’. The second cluster included ‘Mrs. Burns’, ‘Vietnamese’, and ‘Vana holy basil’ cultivars. The third cluster featured sweet basil cultivars, including ‘Clove’, ‘Thai tömzsi’, ‘Fahéj illatú’, ‘Thai hosszú’, and ‘Licorice’, while the final cluster was exclusively composed of ‘Mammoth’ and ‘African basil’ cultivars. In addition, principal component analysis (PCA) revealed relationships between genotypes and quantitative morphological traits (see Figure 2). These two principal components collectively explained 68.99% of the variance, with PC1 accounting for 43.60% and PC2 for 25.39%.

### 2.2. Chemical Attributes of Ocimum Species 

#### 2.2.1. Essential Oil Content and Essential Oil Yield 

This research finding revealed a significant difference (*p* < 0.01) among the genotypes in terms of essential oil production, as presented in Figure 3. The essential oil content (EOC) among the genotypes ranged between 0.4% and 1.7%. The highest EOC was observed in the sweet basil cultivar ‘Mrihani’, followed by ‘Fino Verde’ (1.5%) and ‘African basil’ (1.5%). In contrast, both holy basil genotypes (‘green holy basil’ and ‘Vana holy basil’) had <0.5% essential oil. Clove and ‘African basil’ had the highest EOY (≥1.2 mL/plant). In contrast, ‘Vana holy basil’ had the lowest EOY (0.1 mL/plant). 

#### 2.2.2. Essential Oil Composition 

The essential oil composition among the 15 basil cultivars exhibited remarkable variation, characterized by a predominant presence of either a single compound or a combination of multiple compounds (see Table 3 and Table 4). Over 50 compounds have been identified in each sweet basil (*Ocimum basilicum*) cultivar. Estragole, for instance, was found in higher proportions, exceeding 60%, in the essential oils of ‘Clove’, ‘Mrihani’, ‘Thai hosszú’, ‘Thai tömzsi’, and ‘Vietnamese basil’. However, other sweet basil cultivars showed a mixture of multiple compounds with different ratios. Consequently, ‘Anise’ basil, ‘Fahéj illatú’, and ‘Licorice’ basil exhibited higher ratios of (*E*)-methyl cinnamate and linalool, with percentages of 28.7/12.2%, 34.8/28.1%, and 42.7/19.3%, respectively. Additionally, ‘Anise’ basil also had a significant ratio of estragole (25.1%). Notably, ‘Fino Verde’ and ‘Mammoth’ basil predominantly contained linalool, exceeding 30% in their essential oil composition. The volatile oil of ‘Mrs. Burns’ basil, on the other hand, displayed a composition comprising estragole (21.9%), linalool (19.5%), and citral (12.6%). In addition, more than 60 compounds were detected in both holy basil cultivars. In ‘green holy basil’ of Ethiopian origin, the main essential oil components included *β*-bisabolene (14.1%), linalool (12.1%), estragole (10.1%), and eugenol (7%). However, ‘Vana holy basil’ essential oil was characterized by higher levels of eugenol (27.3%), caryophyllene oxide (14.4%), and (*E*)- *β*-Caryophyllene (6.8%). The ‘Lemon’ basil essential oil was found to contain 55 different compounds, with estragole making up 61.4% and citral, comprising neral and geranial, accounting for 13.9% of the composition. Meanwhile, ‘African basil’ was found to contain 39 different compounds, with higher proportions of monoterpenes, including 20.7% *p*-Cymene and 10.6% *γ*-terpinene, as well as a significant presence of phenylpropanoids, accounting for 42.1% thymol. 

Furthermore, the hierarchical cluster analysis based on the composition of essential oils revealed the presence of four distinct groups (Figure 4). In the first cluster, ‘Anise’, ‘Fahéji illatu’, and ‘Licorice’ basil were grouped due to their higher levels of (*E*)-methyl cinnamate (>28%) and linalool (>12%). In the second cluster, there was a mixture of components with varying ratios. For instance, ‘Fino Verde’ and ‘Mammoth’ basil displayed linalool ratios exceeding 32.7%. Conversely, ‘Thai tömzsi’ and ‘Mrs. Burns’ basil contained higher levels of estragole (>21%) and linalool (19%). Vana holy basil exhibited significant proportions of eugenol (27.3%) and caryophyllene oxide (14.4%) in the second cluster. The ‘African basil’ was categorized in the third cluster, characterized by a higher ratio of thymol (42.1%) and p-cymene (20.7%). The fourth cluster encompassed ‘Clove’, ‘Thai hosszú’, ‘Vietnamese’, ‘Mrihani’, and ‘lemon’ basils, all of which contained over 60% estragole. As indicated in Figure 5, the PCA further provides insight into the relationship between genotypes and essential oil components. The two principal components explained 47.38% of the variance, PC1 explaining 26.80% and PC2 explaining 20.58%.

## 3. Discussion

The *Ocimum* genus represents a notably diverse and valuable medicinal plant, characterized by significant genetic variation, differing morphological traits [29,30], as well as variations in the content and composition of essential oils [29,31,32]. These aromatic essential oils and the nonvolatile compounds found in basil species are used in flavoring, fragrance, cosmetics, aromatherapy, and pharmaceuticals. They are highly regarded for their diverse biological properties, including antimicrobial activity, insecticidal effects, antioxidant capabilities, and numerous therapeutic benefits [13,19,22,24]. Nevertheless, it is important to note that the production, accumulation, and distribution of secondary metabolites are significantly influenced by genetic factors, developmental stages, morphogenetic processes, environmental conditions, and processing techniques [26,27,28]. Accordingly, the morphological traits, essential oil production, and composition of 15 distinct *Ocimum* genotypes belonging to four distinct species of *Ocimum* were evaluated and discussed below. There is considerable variation in morphology among *Ocimum* species, encompassing both vegetative and reproductive growth characteristics. Among these genotypes, African basil exhibits robust growth, with taller plants, broader canopy spreads, and greater biomass production. Comparable growth and yield were also recorded from ‘Clove’, ‘Licorice’, ‘Thai tömzsi’, and ‘Thai hosszú’. In terms of leaf size and weight, ‘Mammoth’ basil exhibits the largest. In contrast, ‘Fino Verde’ basil plants were dwarf types, which were short and small-leafed. Although they have smaller leaves and lower herb yields, they produce the highest number of inflorescences. Earlier studies have demonstrated a significant morphological diversity within *Ocimum* species, a result of factors such as polyploidy, aneuploidy, inter- and intra-specific hybridizations, as well as cultivation and breeding practices [10,29,33,34,35]. As a polymorphic species, *Ocimum gratissimum* exhibits a broader range of morpho-chemical traits, including plant height and fresh herb yield spanning from 46 to 123 cm and 98 to 618.3 g/plant, respectively [5,36,37,38,39], consistent with our own findings. In the case of *Ocimum basilicum* cultivars, previous research has shown variability in height (29.2 to 100 cm), canopy diameter (24.9 to 97.7 cm), and fresh yield (140 to 634 g/plant), influenced by factors such as growing conditions, cultivation seasons, geographical location, and specific cultivars [5,29,30,37,40]. Notably, ‘Fino Verde’ of *O. basilicum* exhibited a height of over 60 cm and a fresh herb yield exceeding 550 g/plant, contrasting with our findings [40]. Furthermore, green holy basil hand height varies from 37.3 to 71.9 cm with a canopy spread of 50 to 67.3 cm [41]. Despite differences in cultivation practices and microclimates, green holy basil is reported to produce between 110 and 361.36 g/plant of fresh biomass [5,39].

The observed variability in essential oil content (EOC) was significant. The ‘Mrihani’ basil exhibited the highest EOC at 1.7%, followed by ‘Fino Verde’ and the African basil varieties. Conversely, ‘Mrs. Burns’, ‘green holy basil’, and ‘Vana holy basil’ displayed the lowest EOC, measuring less than 0.7%. Previous research has suggested that, aside from genetic factors, essential oil content and composition ratios are significantly influenced by growing conditions, growth stage, and processing methods. For instance, EOC in sweet basil cultivars has been reported to range from 0.11% to 3.4% in various studies [42,43,44]. Specifically, ‘Fino Verde’s’ essential oil content has been documented to range from 0.5% to 3.4% in different studies [12,40,45]. In contrast to our findings, the EOC of Anise basil fluctuates between 0.62% and 2.67%, depending on the season and geographical region [12,46]. Similarly, green holy basil was reported to have 0.43% EO [47]. Consistent with our results, previous studies have reported varying essential oil content (EOC) for *O. gratissimum*, ranging from 0.11% to 1.5%. This variability is attributed to factors such as different chemotypes, production seasons, plant parts used, and cultivation regions, which were documented [12,38,48,49]. A vast variety of aromatic compounds are present in the essential oil of basil species, which is recognized for its intricate complexity. The *Ocimum* species of basil include a large number of cultivars and chemotypes, each of which has a distinctive profile of volatile oils. Based on previous studies [27,50,51], this composition can exhibit significant variations due to factors like geographic location, environmental conditions, cultivation techniques, the state of the plant material (fresh or dry), and genetic diversity. Additionally, processes like hybridization that occur both within and between species through cross-pollination, as well as natural evolutionary events, polyploidy, and selective breeding, have all significantly influenced chemical diversification among diverse *Ocimum* species. These phenomena have been documented in various studies [9,14,52]. In line with that, *Ocimum basilicum* has been categorized into four primary chemotypes based on the composition of its essential oils, as outlined by Lawrence et al. [50]. These chemotypes include methyl chavicol (estragole), linalool, methyl eugenol, and methyl cinnamate, with each of them further exhibiting several subgroups. Our observations identified the presence of estragole, methyl cinnamate, linalool, and their combinations in sweet basil cultivars. Our finding further corresponds with the findings of Zeljković et al. [44], who reported substantial levels of linalool (>55%) in Italian (Fino Verde Compatto) and Mammoth basil. It was also documented that linalool (27%) and 1,8-cineole (13%) are the primary compounds in Fino Verde essential oil [45]. Furthermore, Carovic’-Stanko et al. [11] reported Thai basil accessions that were rich in estragole (78.2%) and linalool (46.16%). Earlier research findings have indicated that Anise basil displays a wide range of chemical profiles, with methyl chavicol ranging from 12% to 82.2% and linalool within a range of 30% to 56% [12,42,53]. Additionally, Thai basil has been reported to contain significant amounts of methyl chavicol (20% to 90%) and linalool (>20%) [12,53]. In agreement with our findings, Vieira and Simon [7] and Couto et al. [53] also observed that Mrs. Burns basil oil predominantly consists of linalool (38.3% to 46.1%) and citral (16.6% to 49.56%). Conversely, the composition of other species was characterized by the presence of multiple components. Hence, green holy basil predominantly consists of bisabolene, linalool, estragole, and eugenol. Lemon basil, on the other hand, is primarily composed of estragole and citral. A chemotype of lemon basil from New Guinea was identified, primarily composed of methyl chavicol (estragole) but devoid of citral [7]. Additionally, they noted the existence of two lemon basil accessions with higher citral ratios. Another study highlighted that the principal components of Lemon basil (*O. citriodorum*) volatile oil were estragole and citral [54]. Regarding the chemotaxonomy of *O. gratissimum*, six chemotypes have been reported, including thymol, eugenol, citral, methyl cinnamate, linalool, and geraniol. Consequently, the two *O. gratissimum* cultivars in our study, Vana holy basil, and African basil, belong to the eugenol and thymol chemotypes, respectively. In line with our findings, various authors have also documented that eugenol constitutes the predominant volatile oil compound in Vana holy basil, accounting for more than 50% of the total composition [38,48,54]. Furthermore, thymol and cymene were also reported as primary components in the volatile oil of African basil cultivars [11,55,56]. 

## 4. Materials and Methods 

### 4.1. Experimental Site Description 

The experiment was conducted in 2022 at the Hungarian University of Agriculture and Life Sciences (MATE) experimental field in Budapest-Soroksár. During the investigation, an average air temperature of 15 °C and a rainfall of 234 mm were recorded. Figure 6 illustrates the daily weather conditions, including rainfall and air temperature. Furthermore, Table 5 shows the soil macronutrient and micronutrient concentrations. It had high concentrations of phosphorus (544.43 mg/kg), potassium (177.36 mg/kg), and significant amounts of micronutrients.

### 4.2. Plant Material and Growing Conditions

In this study, we examined 15 distinct basil genotypes, representing four different *Ocimum* species (Figure 7). Among these, eleven genotypes belonged to *Ocimum basilicum* cultivars, two were cultivars of *Ocimum gratissimum*, and the remaining two belonged to *Ocimum citrodora* and *Ocimum sanctum*. The seeds for this experiment were sourced from the Medicinal and Aromatic Plants Department gene bank (MATE), as detailed in Table 6. The germination process involved sowing the seeds in seed trays measuring 27 × 57 cm within a greenhouse during the second week of March. Once healthy seedlings had grown two leaves each, they were transplanted into an open field with a spacing of 40 cm × 40 cm, arranged in 4 rows with 6 plants per row. To ensure scientific rigor, these plantings were replicated three times, following a randomized complete block design (RCBD) arrangement. Throughout the cultivation period, we maintained a watering schedule of three times per week and performed weekly cultivation activities, all without the use of chemical fertilizers or protective chemicals. Harvesting was conducted 10 days after full bloom, and subsequently, fresh herb samples underwent a two-week drying process in a well-ventilated room.

Quantitative phenotypic traits: Eleven morphological features were quantified for characterization. The plant heights (in cm), canopy diameters (in cm), and fresh herb weights (in g/plant) of ten plants were measured during harvest. After allowing the herbs to dry naturally in a well-ventilated room, we recorded their dried weights (in g/plant). For leaf metrics such as leaf length, width, and area, we took the mean of 20 leaves per genotype and two leaves per sample plant from the upper third internode. After acquiring the images of the leaves, image analysis software (ImageJ (version 1.54h), National Institutes of Health, Bethesda, MD, USA) was used to determine the leaf area (in cm^2^/leaf). Ten plants per genotype and three inflorescences per genotype were counted and measured to determine the number of inflorescences and their lengths (cm). Upon adequate seed extraction and drying, 1000 seeds of each genotype were weighed. 

Essential oil content determination: For each treatment, we harvested a total of ten plant samples, which were subsequently dried for two weeks in well-ventilated rooms. After drying, we gathered bulk samples made up of dried leaves and inflorescences (without stems) to measure the amount of essential oils present. Each time, we used 20 g of the dried material from the samples and hydro-distilled it for two hours in a Clevenger-style device with 500 mL of distilled water. This procedure followed the recommendations given in the VII Hungarian Pharmacopoeia [57]. After carefully removing any traces of water that remained in the essential oils, the samples were placed in a sealed vial and kept in a refrigerator set to 4 °C for one day.

Essential oil composition analysis: We employed the gas chromatography-mass spectrometry (GC-MS) method to analyze the composition of the essential oil. The GC analysis was carried out using an Agilent Technologies 6890 N instrument (Agilent Technologies, Inc., Santa Clara, CA, USA), equipped with an HP-5MS capillary column measuring 30 m in length, 0.25 mm in diameter, and featuring a 0.25 μm film thickness. The instrument followed a temperature program that began at an initial temperature of 60 °C with a gradual heating rate of 3 °C per minute until reaching 240 °C, which was maintained for 5 min. The injector and detector temperatures were set at 250 °C. Helium served as the carrier gas, flowing at a constant rate of 1 mL per minute with a split ratio of 30:1 and an injection volume of 0.2 μL (diluted to 1% in n-hexane). The individual compound proportions were expressed as percentages of the total area. The same equipment was utilized for component identification, employing an Agilent Technologies MS 5975 detector (Agilent Technologies, Inc., Santa Clara, CA, USA). The ionization energy was set at 70 eV, and the mass spectra were recorded in full scan mode, generating total ion current (TIC) chromatograms. To calculate linear retention indices (LRI), a mixture of aliphatic hydrocarbons (C9-C23) in n-hexane was injected, applying the generalized equation proposed by Van Den Dool and Kratz [58]. The LRI values and mass spectra were cross-referenced with commercial databases (NIST, Wiley, Hoboken, NJ, USA), Adams [59], and a homemade library mass spectra were built up from data that were obtained from standard (Sigma/Aldrich, St. Louis, MO, USA) pure compounds. The GC samples underwent three repetitions for accuracy.

Data analysis: The morphological and essential oil data of the genotypes were evaluated through a one-way analysis of variance (ANOVA). Before conducting the ANOVA, the data’s normal distribution and homogeneity of variances were verified using Shapiro–Wilk and Levene’s tests, respectively. Significant differences in means were investigated using Tukey’s post hoc test at a significance level of *p* < 0.05. Furthermore, a hierarchical cluster analysis was carried out utilizing the Ward method based on squared Euclidean distance, leading to the creation of a dendrogram. All statistical analyses were executed using IBM SPSS 29 software, while principal components analysis (PCA) and dendrogram generation were performed using Origin Pro 2023b software.

## 5. Conclusions

This research conclusion highlights the remarkable phenotypic and biochemical diversity observed within the fifteen basil genotypes preserved in the department’s gene bank. There is a wide range of diversity in morphological traits, essential oil production, and composition. Notably, certain genotypes, including African basil, Clove, Licorice, Thai tömzsi, and Thai hosszú, exhibited robust morphological growth. Comparatively, Mrihani, Fino Verde, and African basil varieties produced higher levels of essential oil. Moreover, the investigation revealed a wide range of intra- and inter-specific chemical diversity in the essential oil compositions of *Ocimum* taxa, including estragole, thymol, methyl cinnamate, linalool, eugenol, and various mixed compositions. This comprehensive analysis significantly contributes to our understanding of their inherent diversity in this growing region (Hungary). 

## Figures and Tables

**Figure 1 plants-13-00064-f001:**
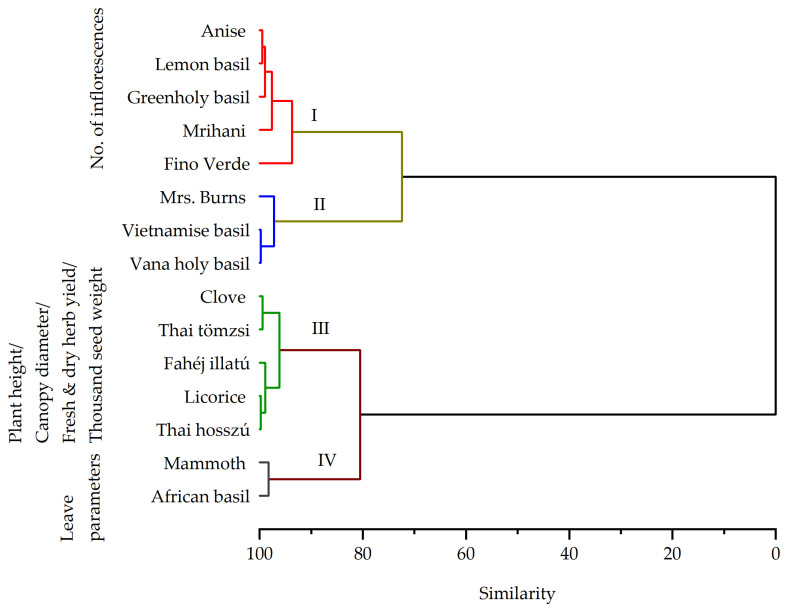
Morphological cluster analysis among basil genotypes.

**Figure 2 plants-13-00064-f002:**
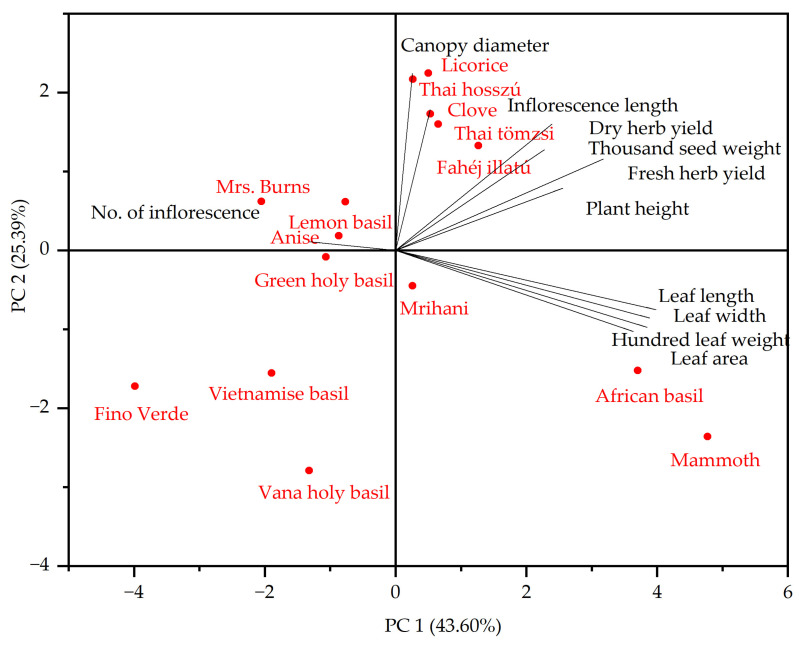
Principal component analysis plot for morphological traits among the 15 genotypes.

**Figure 3 plants-13-00064-f003:**
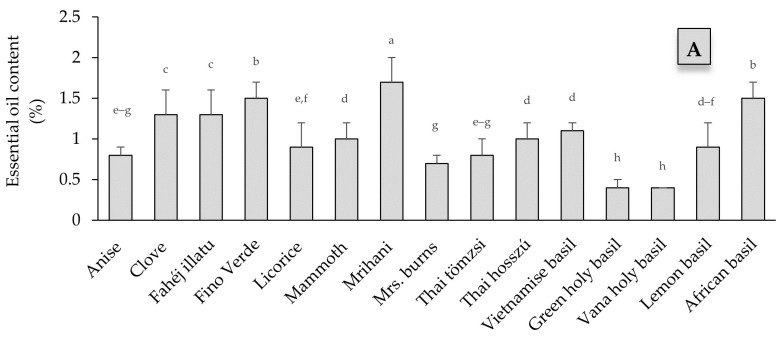
Variability in essential oil production of 15 *Ocimum* genotypes (**A**) essential oil content and (**B**) essential oil yield. Values are presented as Mean ± SD; Different letters indicate significantly different means.

**Figure 4 plants-13-00064-f004:**
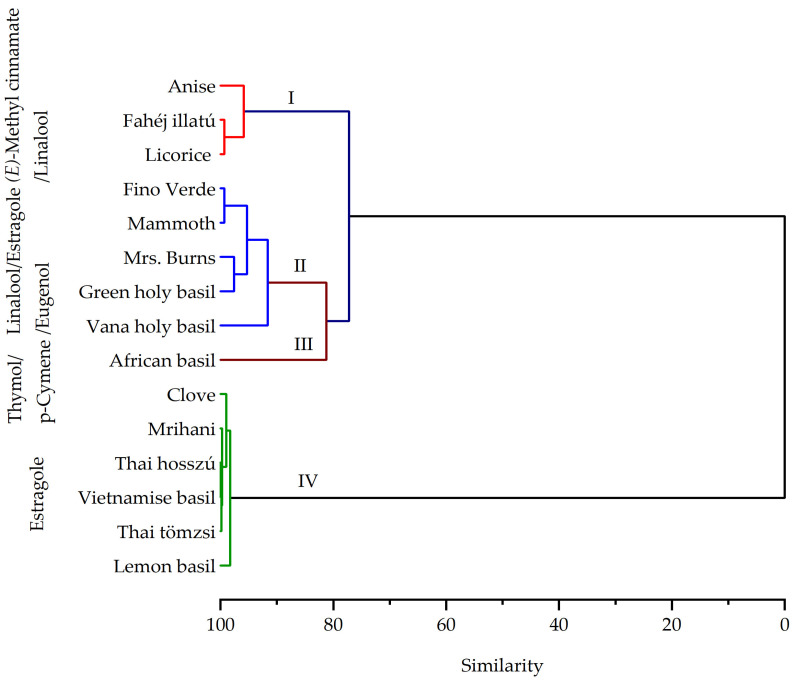
Essential oil composition cluster analysis.

**Figure 5 plants-13-00064-f005:**
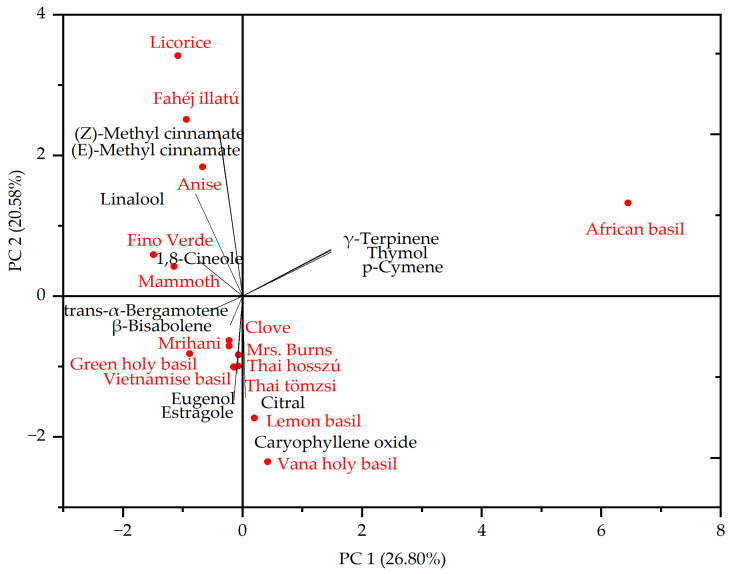
Principal component plot analysis of volatile oil composition among the 15 genotypes.

**Figure 6 plants-13-00064-f006:**
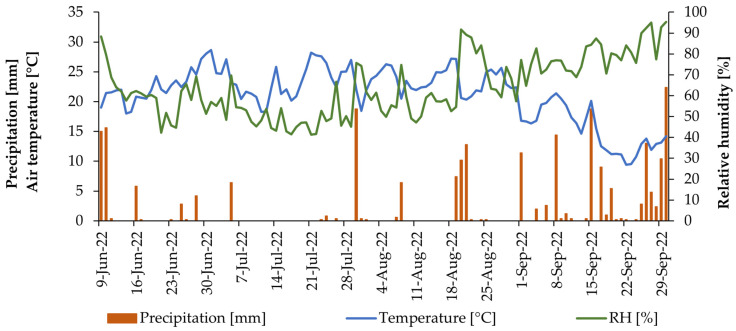
Weather conditions during the experimental period (June to September 2022).

**Figure 7 plants-13-00064-f007:**
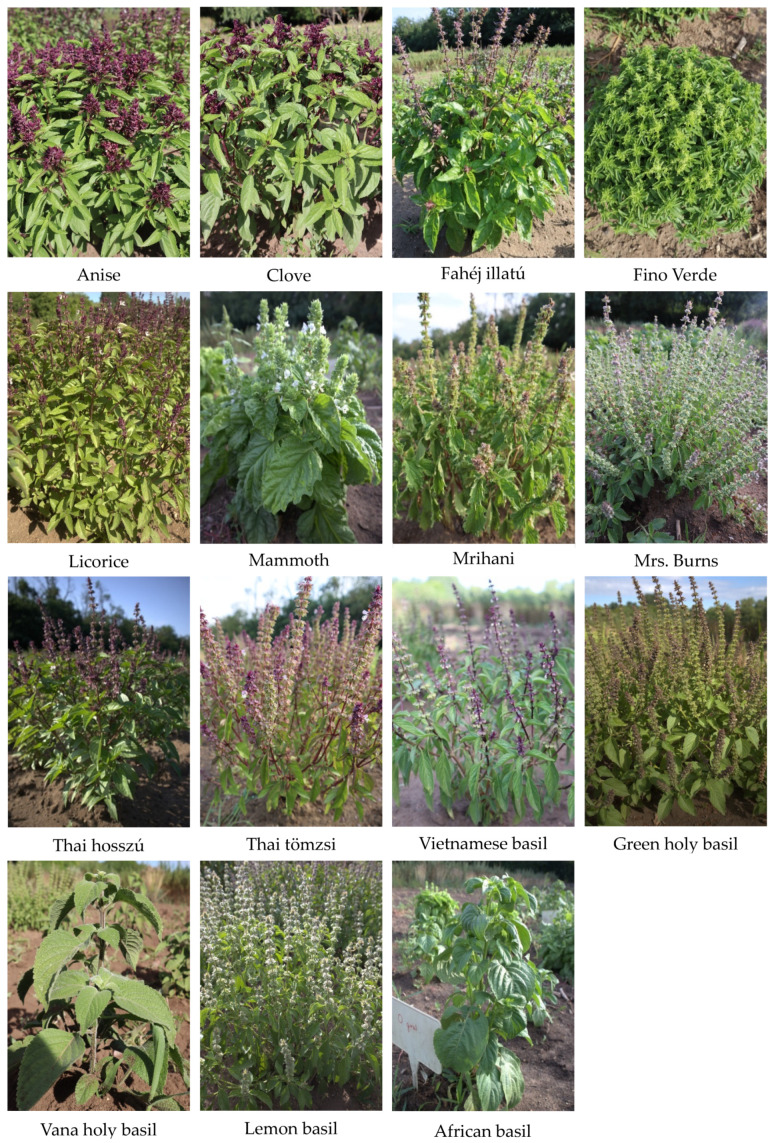
Morphology of the 15 *Ocimum* genotypes.

**Table 1 plants-13-00064-t001:** Growth parameters among *Ocimum* species.

Genotypes	PH (cm)	CD (cm)	LL (cm)	LW (cm)	LA (cm^2^)	HLW (g)
Anise	47.7 ± 3.9 ^c,d^	50.0 ± 2.4 ^a,b^	6.2 ± 0.3 ^e,f^	2.3 ± 0.2 ^i,j^	7.7 ± 1.8 ^e–g^	31.7 ± 2.6 ^g^
Clove	43.0 ± 2.2 ^d,e^	52.5 ± 6.3 ^a^	6.1 ± 0.5 ^e–g^	3.1 ± 0.2 ^f–h^	10.4 ± 2.0 ^d–f^	33.4 ± 3.1 ^g^
Fahéj illatú	50.7 ± 2.8 ^a–c^	47.8 ± 3.4 ^a,b^	7.6 ± 0.7 ^c^	4.3 ± 0.4 ^c,d^	14.8 ± 4.8 ^d^	70.0 ± 5.1 ^c^
Fino Verde	23.0 ± 3.0 ^g^	32.5 ± 4.7 ^e,f^	1.8 ± 0.3 ^i^	0.9 ± 0.1 ^k^	1.4 ± 0.3 ^g^	3.5 ± 0.6 ^j^
Licorice	54.3 ± 4.4 ^a,b^	52.5 ± 4.6 ^a^	5.8 ± 0.4 ^e–g^	3.3 ± 0.3 ^f,g^	9.9 ± 2.2 ^d–f^	30.9 ± 3.2 ^g^
Mammoth	43.3 ± 3.2 ^d,e^	38.3 ± 4.7 ^c–e^	12.4 ± 1.3 ^a^	8.6 ± 1.1 ^a^	52.0 ± 1.2 ^a^	214.0 ± 17.1 ^a^
Mrihani	44.7 ± 3.8 ^c,d^	44.8 ± 5.1 ^a–c^	7.2 ± 0.6 ^c,d^	4.9 ± 0.6 ^c^	16.0 ± 3.4 ^d^	62.3 ± 3.9 ^d^
Mrs. Burns	47.2 ± 5.0 ^c,d^	45.2 ± 4.5 ^a–c^	4.1 ± 0.6 h	1.9 ± 0.3 ^j^	4.5 ± 1.3 ^f,g^	13.5 ± 2.3 ^i^
Thai tömzsi	50.8 ± 1.5 ^a–c^	47.3 ± 4.8 ^a–c^	6.1 ± 0.8 ^e–g^	3.1 ± 0.2 ^f–h^	8.2 ± 1.2 ^e,f^	42.4 ± 4.3 ^f^
Thai hosszú	50.8 ± 2.8 ^a–c^	50.5 ± 3.3 ^a,b^	5.2 ± 0.3 ^g^	2.7 ± 0.4 ^g–i^	7.7 ± 1.9 ^e–g^	53.8 ± 4.5 ^e^
Vietnamese	37.8 ± 1.5 ^e,f^	34.3 ± 2.7 ^d–f^	5.4 ± 0.5 ^f,g^	2.6 ± 0.2 ^h,i^	6.8 ± 1.8 ^e–g^	21.0 ± 2.8 ^h,i^
Green holy basil	43.7 ± 3.3 ^d,e^	47.8 ± 8.8 ^a,b^	5.4 ± 0.2 ^f,g^	3.4 ± 0.1 ^e,f^	9.9 ± 0.7 ^d–f^	34.9 ± 4.4 ^g^
Vana holy basil	48.5 ± 4.7 ^b–d^	27.2 ± 5.3 ^f^	6.6 ± 0.3 ^d,e^	4.0 ± 0.1 ^d,e^	12.8 ± 2.9 ^d,e^	27.3 ± 2.7 ^g,h^
Lemon basil	34.0 ± 3.1 ^f^	42.5 ± 7.1 ^b–d^	4.1 ± 0.3 ^h^	2.0 ± 0.3 ^j^	22.6 ± 2.6 ^c^	18.5 ± 1.9 ^i^
African basil	56.3 ± 4.7 ^a^	32.8 ± 2.4 ^e,f^	10.3 ± 1.4 ^b^	6.9 ± 0.7 ^b^	36.6 ± 9.6 ^b^	148.0 ± 12.8 ^b^

Values are presented as Mean ± SD; PH: plant height, CD: canopy diameter, LL: leaf length, LW: leaf width, LA: leaf area, HLW: hundred leaf weight, Different letters indicate significantly different means.

**Table 2 plants-13-00064-t002:** Variation in reproductive traits and yield among *Ocimum* species.

Genotypes	IL (cm)	NI	TSW (g)	FHW (g/Plant)	DHW (g/Plant)
Anise	13.5 ± 1.0 ^f,g^	74.2 ± 13.4 ^c,d^	1.1 ± 0.1 ^d^	249.2 ± 61.1 ^a–d^	60.0 ± 16.2 ^a–d^
Clove	16.5 ± 1.8 ^e–g^	66.5 ± 4.4 ^c–e^	1.7 ± 0.1 ^a^	403.0 ± 123.2 ^a^	95.2 ± 30.7 ^a^
Fahéj illatú	29.0 ± 4.5 ^a^	60.3 ± 27.3d ^e^	1.6 ± 0.1 ^a,b^	331.3 ± 133.0 ^a,b^	72.3 ± 24.5 ^a–c^
Fino Verde	6.5 ± 1.8 ^g^	152.7 ± 31.4 ^a^	0.9 ± 0.0 ^e^	189.7 ± 49.1 ^b–e^	44.3 ± 10.5 ^b–e^
Licorice	25.7 ± 2.1 ^a–c^	91.0 ± 13.2 ^b–d^	1.5 ± 0.1 ^a–c^	350.7 ± 61.7 ^a,b^	91.8 ± 31.0 ^a^
Mammoth	14.8 ± 14.8 ^f,g^	70.3 ± 6.3 ^c,d^	1.5 ± 0.0 ^a–c^	396.7 ± 91.6 ^a^	70.2 ± 17.8 ^a–c^
Mrihani	19.7 ± 2.4 ^d,e^	68.8 ± 11.5 ^c–e^	1.4 ± 0.0 ^b,c^	212.7 ± 21.6 ^b–e^	49.3 ± 14.6 ^b–e^
Mrs. Burns	28.8 ± 1.9 ^a^	94.5 ± 24.2 ^b,c^	1.4 ± 0.0 ^b,c^	128.3 ± 82.1 ^c–e^	36.0 ± 24.1 ^c–e^
Thai tömzsi	24.3 ± 2.5 ^b,c^	31.3 ± 3.0 ^f^	1.4 ± 0.1 ^b,c^	408.7 ± 116.3 ^a^	82.0 ± 22.7 ^a,b^
Thai hosszú	27.2 ± 2.3 ^a,b^	90.3 ± 18.7 ^b–d^	1.6 ± 0.0 ^a–c^	355.0 ± 85.4 ^a,b^	82.7 ± 17.8 ^a,b^
Vietnamese basil	18.5 ± 2.2 ^e,f^	22.7 ± 2.9 ^f^	1.3 ± 0.1 ^c,d^	82.0 ± 17.2 ^e^	18.5 ± 4.7 ^e^
Green holy basil	17.3 ± 1.2 ^e–g^	108.5 ± 28.7 ^b^	0.5 ± 0.0 ^f^	284.7 ± 117.4 ^a–c^	76.0 ± 24.2 ^a–c^
Vana holy basil	14.7 ± 1.2 ^f,g^	39.0 ± 6.9 ^e,f^	1.1 ± 0.0 ^d^	94.8 ± 11.3d ^e^	24.2 ± 4.7 ^d,e^
Lemon basil	22.8 ± 2.8 ^c,d^	64.2 ± 8.1 ^c–e^	1.5 ± 0.0 ^a–c^	268.7 ± 132.9	76.17 ± 35.8 ^a–c^
African basil	14.7 ± 1.3 ^f,g^	72.7 ± 6.3 ^c,d^	1.3 ± 0.1 ^c,d^	408.3 ± 39.1 ^a^	84.3 ± 7.0 ^a,b^

Values are presented as Mean ± SD; IL: Inflorescence length; NI: Number of inflorescences; TSW: thousand seed weight; FHW: fresh herb weight; DHW: dry herb weight; Different letters indicate significantly different means.

**Table 3 plants-13-00064-t003:** Essential oil compositions vary among sweet basil cultivars.

Component	RT	LRI	Anise	Clove	Fahéj Illatú	Fino Verde	Licorice	Mammoth	Mrihani	Mrs. Burns	Thai Hosszú	Thai Tömzsi	Vietnamese Basil
β-Myrcene			0.1	0.2	0.1	0.7	0.1	0.3	1.2	0.1	0.4	0.4	0.7
1,8-Cineole	8.44	1034	3.6	2.6	2.4	12.1	5.3	4.2	8.2	0.3	1.9	5.0	2.6
(*E*)-Ocimene	8.96	1046	0.4	2.7	0.2	0.2	0.2	0.1	0.2	0.3	1.4	1.0	1.6
Linalool	10.88	1097	12.2	11.8	28.1	32.7	19.3	32.8	2.5	19.5	2.3	1.8	2.5
Camphor	12.69	1144	1.0	0.5	0.4	1.8	0.8	0.3	0.8	0.0	1.9	1.2	0.3
Terpinen-4-ol	13.97	1175	1.4	0.6	0.9	0.1	2.0	5.4	0.1	1.7	0.5	0.1	0.1
Estragole	14.83	1196	25.1	62.2	1.5	0.4	1.6	7.3	66.9	21.9	65.3	64.0	67.4
Nerol	16.14	1227	0.0							6.6			
Neral (citral b)	16.72	1238	0.0							5.5			
Geranial (citral a)	18.00	1268	0.0		0.0					7.1			
Isobornyl acetate	18.52	1284	1.0	0.3	0.3	2.2	0.6	0.3	1.4	0.0	0.5	0.7	2.9
(*Z*)-Methyl cinnamate	19.34	1299	5.4		4.7		8.3						
Eugenol	21.49	1361			2.0	8.3	0.0	1.8		0.1			
(*E*)-Methyl cinnamate	22.64	1394	28.7		34.8		42.7	0.2			0.2	0.2	0.3
β-Elemene	22.92	1391		0.6		2.9		2.2	0.8	1.6	0.7	0.6	0.5
Methyl eugenol	23.48	1411	0.4	1.1	0.1	0.4	0.1	0.1	0.7	0.1	4.9	4.6	0.8
(*E*)-β-Caryophyllene	24.00	1420	0.3	0.2	0.5	0.3	0.2	0.4	0.7	2.7	0.2	0.4	0.3
*trans*-α-Bergamotene	24.69	1437	1.1	2.1	0.9	4.0		8.4		1.2	3.4	1.1	3.4
Germacrene D	26.49	1482	1.5	1.3	2.4	2.5	1.5	3.8	0.5	2.0	0.8	0.4	0.3
α-Bulnesene	27.48	1506	1.0	0.4	1.5	2.9	1.2	2.0	0.5	1.1	0.6	0.6	0.3
*cis*-γ-Cadinene	27.8o	1515	1.2	1.1	1.7		1.1	2.9		1.1	1.0	0.9	1.1
*cis*-α-Bisabolene	28.95	1544								2.7			
Maaliol	29.83	1580	0.9				0.4	1.8			0.1		0.1
Caryophyllene oxide	30.46	1590	0.3		0.1	0.1	0.1	0.1		2.6	0.3	1.0	0.5
τ-Cadinol	32.62	1644	4.4	4.6	6.1	8.4	4.7	10.2	6.7	4.6	4.0	4.9	3.5
others (<1%)			7.1	6.2	9.1	18.1	7.6	12.9	7.9	13.7	6.2	7.3	10.6
Total identified (%)			97.1	98.5	97.8	98.1	97.8	97.5	99.1	96.5	96.6	96.2	99.8
Monoterpenes			1.4	4.0	0.9	3.4	1.6	2.5	3.5	0.9	2.7	2.6	3.3
Oxygenated monoterpenes			20.7	17.7	34.4	52.3	30.0	45.0	13.4	46.3	7.84	10.7	12.0
Sesquiterpenes			7.1	7.2	9.5	20.7	5.7	25.0	5.5	13.5	8.1	5.3	7.4
Oxygenated sesquiterpenes			8.2	6.2	9.5	11.9	7.7	15.1	8.1	11.6	7.4	9.3	6.7
Phenylpropanes			59.8	63.6	43.5	10.0	52.8	9.8	68.8	24.7	70.6	69.2	68.8

RT—retention time. LRI—linear retention index relative to C9-C23 n-alkanes on an HP-5MS capillary column.

**Table 4 plants-13-00064-t004:** Essential oil composition varies among the three *Ocimum* species.

Component	RT	LRI	Green Holy Basil	Vana Holy Basil	Lemon Basil	African Basil
*β*-myrcene	7.09	994	0.2			3.1
α-Terpinene	7.93	1018		0.1		2.7
*p*-Cymene	8.17	1028				20.7
γ-Terpinene	9.36	1056				10.6
1,8-Cineole	8.44	1034	8.3		0.7	0.1
(*E*)-Ocimene	8.96	1046	0.7	2.6	0.3	
*cis*-Sabinene hydrate	9.68	10.68	0.6	1.4		1.2
Linalool	10.88	1097	12.1	2.8	3.5	0.3
Terpinen-4-ol	13.97	1175	0.2	0.6		1.1
Estragole	14.83	1196	10.1	0.4	61.4	0.8
Isobornyl acetate	18.52	1284		1.5	0.1	
Thymol	18.81	1290				42.1
Nerol	16.14	1227	0.9		2.1	
Neral (citral b)	16.72	1238	0.8		6.1	
Geranial (citral a)	18.00	1268	1.2		7.8	
Eugenol	21.49	1361	7.0	27.3	0.1	
α-Copaene	22.25	1377	0.2	2.7	0.1	
(*E*)- *β*-Caryophyllene	24.00	1420	1.1	6.8	1.1	2.2
*trans*-α-Bergamotene	24.69	1437	2.3	0.0	0.9	
α-Humulene	25.38	1454	1.2	0.7	0.3	0.3
Germacrene D	26.49	1482	0.6	5.4	0.2	
β-Bisabolene	27.63	1508	14.1		0.1	
*cis*-α-Bisabolene	28.95	1544	4.9		1.3	
Caryophyllene oxide	30.46	1590	3.2	14.4	2.6	
Humulene epoxide II	31.44	1615	2.4	1.0	0.3	
τ-Cadinol	32.62	1644	2.7	0.4	0.6	
*3*-*iso*-Thujopsanone	33.00	1653		1.8		
Cubenol	33.13	1658		2.9		
others (<1%)			16.0	23.5	7.9	11.9
Total identified (%)			90.8	96.3	95.5	97.1
Monoterpenes			3.8	0.7	42.7	42.7
Oxygenated monoterpenes			28.1	9.0	22.8	5.7
Sesquiterpenes			27.3	14.2	4.3	4.3
Oxygenated sesquiterpenes			15.3	34.7	4.6	0.3
Phenylpropanes			18.6	34.8	63.4	45.3

RT—retention time. LRI—linear retention index relative to C9-C23 n-alkanes on an HP-5MS capillary column.

**Table 5 plants-13-00064-t005:** Soil characteristics of the experimental field.

pH H_2_O	Humus %	K_A_	NO_3_-N mg/kg	P_2_O_5_ mg/kg	K_2_O mg/kg	Na mg/kg	Mg mg/kg	Mn mg/kg	Zn mg/kg	Cu mg/kg	SO_4_ mg/kg
7.58	1.70	<25	11.36	544.43	177.36	30.11	377.93	58.48	4.90	2.46	71.58

**Table 6 plants-13-00064-t006:** List of basil species, cultivar name, and origin of the 15 basil accessions.

Accession No.	Species	Cultivar/Common Name	Sources
LAMIOCI60	*O. basilicum*	Anise	Department gene bank
LAMIOCI61	*O. basilicum*	Clove	Department gene bank
LAMIOCI62	*O. basilicum*	Fahéj illatú	Department gene bank
LAMIOCI63	*O. basilicum*	Fino Verde	Department gene bank
LAMIOCI64	*O. basilicum*	Licorice	Department gene bank
LAMIOCI65	*O. basilicum*	Mammoth	Department gene bank
LAMIOCI66	*O. basilicum*	Mrihani	Department gene bank
LAMIOCI67	*O. basilicum*	Mrs. Burns	Department gene bank
LAMIOCI68	*O. basilicum*	Thai tömzsi	Department gene bank
LAMIOCI69	*O. basilicum*	Thai hosszú	Department gene bank
LAMIOCI70	*O. basilicum*	Vietnamese basil	Department gene bank
LAMIOCI55	*O. sanctum*	Green holy basil	Department gene bank
LAMIOCI75	*O. citrodora*	Lemon basil	Department gene bank
LAMIOCI77	*O. gratissimum*	Vana holy basil	Department gene bank
LAMIOCI78	*O. gratissimum*	African basil	Department gene bank

## Data Availability

The data is contained within the manuscript.

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
