# Peer review of "Phenotypic Variations and Bioactive Constituents among Selected Ocimum Species"

_plants, 2023, doi:10.3390/plants13010064_

Round 1

Reviewer 1 Report

Comments and Suggestions for Authors

The importance on detecting bioactive constituents of essential oils from different Ocimum Species should be further discussed.

Author Response

Dear reviewer

I appreciate your constructive feedback. Based on your suggestions I have amended the manuscript. The changes are indicated with blue color. Let me know if further improvements are required.

Thank you

Reviewer 2 Report

Comments and Suggestions for Authors

In the current manuscript, Mulugeta and co-authors studied the diversity in the morphological characteristics and essential oil composition among basil genotypes. Overall work presented in the manuscript is interesting. Experiments are planned and executed well. Results are also well discussed. I will recommend manuscript be considered for publication after minor revision.

Authors can consider the following comments while improving the present draft of a manuscript,

1.      Please provide the full scientific name of the plants in the abstract, instead of an abbreviation.

2.      Line 44: Please provide binomial names for plant species.

3.      Line 47-49: Rewrite the sentence  

4.      Instead of “morphology”, the word “phenotype” would be much more appropriate.

5.      In some places, font size is not uniform. Please correct it. 

Comments on the Quality of English Language

English is okay.

Author Response

Dear reviewer,

I appreciate and value your insightful input. I have revised the manuscript according to your recommendations, and the modifications are highlighted in blue. Please inform me if additional enhancements are necessary.

Thank you.

Reviewer 3 Report

Comments and Suggestions for Authors

The manuscript entitled "Variation in Morphology and Bioactive Constituents among 2 Selected Ocimum Species" is brief, clear and well written pepper based on characterization of very well known plant species.

In the article should highlight the innovative nature of the research.

Why the authors  performed experiments on such a well-known species, the biodiversity of which has also been known for decades. Otherwise, the manuscript does not cover any new topics. It is a study of the so-called reproductive, based on known techniques on very well-known plant raw material.

 There are also some points need to be corrected:

 - the table 5 and 6 lacks uniformity in writing the names of compounds. Sometimes with a capital letter, sometimes with a lowercase letter...

 - iso, E, Z, neo etc. - please italicize

- methyl cinnamate <(Z)- and methyl cinnamate (E)-  incorrect spelling of the name of the compound

- metil-eugenol   - mistake

trans-ß-Caryophyllene  -  - it's better to use this way of writting: (E)- ß-Caryophyllene

germacrene d - it is better to write Germacrene D

(E)- ß caryophyllene oxide instead of  caryophyllene oxide

tau-cadinol - "tau" should be as Latin letter.

Add a line to the table  5 "Total identified"

Table 5 and Table 6 - Other (<1%) : - there are many compounds not identified. In one case even 20.6% and 25.4%. It is very high level, especially the Basil EOs are already known and very extensive explored. The authors sholud explain if these others were identified either the others are the sum of unidentified volatiles occurred below 1% in EO.

The discussion based on types of morphological categories and genotypes should be more detailed.

line 228 - different size of letters

Table 6: terpinene <α-> , cymene <ο-> , bisabolene <β-> , thujopsanone <3-iso->   -  incorrect way of writing the name, it look like copy/paste from program for the interpretation of mass spectra

I would like to ask the authors what does it mean " The researchers propose that future investigations should ....and phenol composition to gain further insights into this subject"

Author Response

Dear reviewer,

I appreciate and value your insightful input. I have revised the manuscript according to your recommendations, and the modifications are highlighted in blue. Please inform me if additional enhancements are necessary.

Specific answers 

Why the authors performed experiments on such a well-known species, the biodiversity of which has also been known for decades. Otherwise, the manuscript does not cover any new topics. It is a study of the so-called reproductive, based on known techniques on very well-known plant raw material.

Generally, there exists a misconception surrounding the Ocimum genus. And also the taxonomy is very complicated. Sweet basil, being a highly significant species within this genus, has been thoroughly studied. However, even for this species, the vast number of varieties and cultivars complicates comparisons. While numerous publications focus on individual cultivars, only a handful undertake a comprehensive comparison of multiple cultivars. Other Ocimum species, in general, lack thorough investigation. Nevertheless, the diversity present in holy basil and African basil is comparable to that of sweet basil. The strength of this current publication lies in the comparison of 15 accessions under same environmental conditions. Moreover, these cultivars are cultivated in this region, and our institute offers practical information and consulting services related to medicinal and aromatic plants for growers. In addition, the chemical composition of basil (also other MAP) are highly influenced several factors including growing regions.

Regarding the identified components

We were able to identify more than 90% of the components, denoted as total identified compounds (%). Given that each basil cultivar contained more than 50 components, our decision was to specifically report components with ratios exceeding 1%. The remaining components were reported as <1%.

I would like to ask the authors what does it mean " The researchers propose that future investigations should ....and phenol composition to gain further insights into this subject"

In our experience, there is limited literature on the polyphenol profile of Ocimum species. The existing evidence predominantly focuses on a few species, primarily sweet basils. Considering the extensive diversity of basil species and their various biological activities, it is crucial to investigate their phenolic acids, flavonoids, and anthocyanins in addition to their essential oil composition.

Thank you.

Round 2

Reviewer 3 Report

Comments and Suggestions for Authors

Dear Author,

thank you for your response.

I have still some comments to your manuscript.

Table 5 and 6:

wrong name of compound: estragol instead of estragle

terpinen-4-ol - "ol" not italicize

Iso-bornyl acetate - should be Izobornyl acetate

trans-α-bergamotene --> should be trans- α-Bergamotene

Tau-cadino -tau  should be as a Greek letter

Table 5:

why there is a space between iso-bornyl acetate and Z-methyl cinnameta?

cis-γ-Cadinene - please check the correctness of the isomer type. Is it correct to use the term cis in the case of gamma-cadinene?

The author wrote " LRI – linear retention index relative to C9-C23 n-alkanes on an HP-5MS capil- 200

lary column".

In Table 5 there is RI not LRI. Is it experimental or literature retention index? please complete.

Table 6:

I wrote in an earlier review to ask the authors to consider the correctness of the common names of the identified compounds. There are still errors e.g. for:  Sabinene hydrate <cis->.

I should be written as cis-Sabinene hydrate

α-humulene - from capital letter: α-Humulene

Cis-α-Bisabolene should be writtes as: cis-α-Bisabolene

Table 5 and 6:

Cymene isomers are characterized by similar MS spectra. On the HP-5 column, p-cymene has a retention index more similar to the suggested as o-cymene. Additionally, I suggest checking the available literature which isomer of cymene dominates in basil essential oils. According to my knowledge, it is p-cymene and not o-cymene.

O-cymene is the first to elute from the HP-5 column, the next isomer is m-cymene and the third isomer is p-cymene - the most popular in plant essential oils.

Author Response

Dear reviewer,

We appreciate your feedback and constructive suggestions. Based on your comments, we have made revisions to the manuscript. Additionally, we would like to address some of your inquiries:

  1. Regarding compound naming, our primary reference is Admas' book.

  2. The linear retention index is based on experimental data, not the literature retention index.

  3. About the Cymene isomers, we re-evaluated our data and engaged in author discussions. Despite our initial belief in o-Cymene based on retention time, retention index, and libraries, we acknowledge that the majority of the literature reports p-Cymene. Consequently, we have decided to align with the literature and your suggestion, modifying it to p-Cymene in the manuscript.

  4. Considering the slight soil gradient of the experimental field and its cultivation history, we opted for a randomized complete block design (RCBD) with three block replications.

Once again, thank you for your valuable input.

Best regards,